# Giant anisotropic thermal expansion actuated by thermodynamically assisted reorientation of imidazoliums in a single crystal

Zi-Shuo Yao [1,5]*, Hanxi Guan[2,5], Yoshihito Shiota [3], Chun-Ting He[4], Xiao-Lei Wang[1], Shu-Qi Wu[3], Xiaoyan Zheng [1], Sheng-Qun Su[3], Kazunari Yoshizawa[3], Xueqian Kong [2], Osamu Sato[3]* & Jun Tao[1]*

Materials demonstrating unusual large positive and negative thermal expansion are fascinating for their potential applications as high-precision microscale actuators and thermal expansion compensators for normal solids. However, manipulating molecular motion to execute huge thermal expansion of materials remains a formidable challenge. Here, we report a single-crystal Cu(II) complex exhibiting giant thermal expansion actuated by collective reorientation of imidazoliums. The circular molecular cations, which are rotationally disordered at a high temperature and statically ordered at a low temperature, demonstrate significant reorientation in the molecular planes. Such atypical molecular motion, revealed by variable-temperature single crystal X-ray diffraction and solid-state NMR analyses, drives an exceptionally large positive thermal expansion and a negative thermal expansion in a perpendicular direction of the crystal. The consequent large shape change (~10%) of bulk material, with remarkable durability, suggests that this complex is a strong candidate as a microscale thermal actuating material.

[1] Key Laboratory of Cluster Science of Ministry of Education, School of Chemistry and Chemical Engineering, Beijing Institute of Technology, 100081 Beijing, People's Republic of China. [2] Center for Chemistry of High-Performance & Novel Materials, Department of Chemistry, Zhejiang University, 310027 Hangzhou, People's Republic of China. [3] Institute for Materials Chemistry and Engineering, Kyushu University, 744 Motooka, 819-0395 Nishi-ku, Fukuoka, Japan. [4] MOE Key Laboratory of Functional Small Organic Molecule, College of Chemistry and Chemical Engineering, Jiangxi Normal University, 330022 Nanchang, People's Republic of China. [5] These authors contributed equally: Zi-Shuo Yao, Hanxi Guan. *email: zishuoyao@bit.edu.cn; sato@cm.kyushu-u.ac.jp; taojun@bit.edu.cn

Thermal expansion is a fundamental physical property of materials. In typical solid-state materials, the distance between constituent atoms increases upon heating, resulting in a moderate positive thermal expansion (PTE) with a thermal expansion coefficient ($\alpha$) of $<20 \times 10^{-6}\,K^{-1}$. However, some crystalline materials exhibit unusually anisotropic thermal expansion, i.e., the lattice of the material expands remarkably in one direction ($\alpha > 100 \times 10^{-6}\,K^{-1}$) and contracts in another direction ($\alpha < 0 \times 10^{-6}\,K^{-1}$) as temperature increases[1–6]. Large PTE and negative thermal expansion (NTE) properties are not only intriguing for scientific research but also promising for advancing technologies such as high-precision thermal actuators and thermal expansion compensators for normal solids[4,7–14].

To achieve unusual thermal expansion properties, crystalline materials with diverse flexible structures have been explored. Typical examples include metal-organic frameworks that possess wine-rack structures, wherein large uniaxial PTE could occur in one direction with concomitant NTE in the perpendicular direction as a consequence of sideways contraction of the rhombic structure[1,2,15–18]. Unusual thermal expansion can also be induced by the collective significant reorientation of non-spherical molecules[4,6,19–21]. However, the significantly large molecular reorientation is usually restricted due to intermolecular interactions and steric effects in solid-state materials. One promising strategy is to introduce a special entropy-reservoir unit that can perform molecular reorientation by surmounting the energy barrier that prohibits the molecular motion through entropy gain from molecular thermodynamics. We have previously discovered that oxalate anions ($ox^{2-}$) can perform atypical 90° rotation accompanied with a rotationally disordered–statically ordered structural variation in a single crystal of Ni(II) complex[19]. The dramatic reorientation of $ox^{2-}$, which is dominated by competition between molecular thermodynamic rotation and subtle intermolecular interactions, leads to a moderate expansion/contraction (4.9%) in the shape of a single crystal. Thermodynamic rotation is also observed in circular molecules[22–25]; hence, in this study, we focus on circular molecules as an entropy-reservoir unit and envisage that the orientation of molecular planes could be modulated by thermodynamic rotation of circular molecules. If the constituent circular molecules can perform significant reorientation as the temperature changes, such a molecular motion potentially induce a significant variation in the lattice length.

Here, we report an organic–inorganic hybrid Cu(II) complex, bis(imidazolium) tetrachlorocuprate, $(Himd)_2[CuCl_4]$ (1), which exhibits exceptionally large PTE and NTE induced by a thermodynamically assisted reorientation of imidazoliums. In the crystal, the molecular complex is equipped with circular protonated imidazole ($Himd^+$) cations. The cation, which is ordered at low temperature, gradually becomes rotationally disordered upon heating. Due to the synergetic motion of the molecular cations and the counter anions, the plane of $Himd^+$ demonstrates a ca. 30° reorientation as the temperature increases from 123 to 393 K. The collective reorientation of $Himd^+$ cations actuates extremely large linear positive and negative anisotropic thermal expansion of the material. Due to the highly stable crystalline lattice, the molecular level structural variation is effectively transmitted to a macroscopic linear contraction or expansion of the bulk single crystal (ca. 10%), and can be repeatedly observed without noticeable deterioration in the crystal quality.

## Results

**Structural characterization.** Single crystals of 1 can be readily obtained by slow evaporation of a methanol solution containing $CuCl_2 \cdot 4H_2O$ and imidazolium chloride in a stoichiometric ratio.

In the crystal, the circular protonated imidazole cations are employed as entropy-reservoir unit. Single-crystal X-ray diffraction (SCXRD) analysis reveals that complex 1 crystallizes in a triclinic $P-1$ space group at 123 K[26]. In the low-temperature phase (LTP), the asymmetric unit comprises four $Himd^+$ cations and two discrete $CuCl_4^{2-}$ dianions, where the Cu(II) ion is coordinated in a heavily distorted tetrahedral geometry by four chloride anions with Cl–Cu–Cl angles in the range of 95.58(5)°–137.18(5)° (Fig. 1, Supplementary Fig. 1). The complex dianions, $CuCl_4^{2-}$, which are arranged in layers along the crystallographic (010) face, are connected and supported by the interlayer $Himd^+$ cations through weak N–H···Cl and C–H···Cl hydrogen bonds and ionic interactions. Based on the hydrogen interaction motifs, the layers of $Himd^+$ cations can be classified into two groups: (1) layer A, in which two $Himd^+$ cations are surrounded by four $CuCl_4^{2-}$ dianions, forming a rhomboid grid (Supplementary Fig. 2a), and (2) layer B, where $Himd^+$ cations linked by $CuCl_4^{2-}$ dianions produce one-dimensional zig-zag chains parallel to the crystallographic $a$ direction (Supplementary Fig. 2b). In this LTP, the $Himd^+$ cations are statically ordered. As shown in Fig. 1a and Supplementary Fig. 1a, the dihedral angles between the molecular planes and crystallographic (010) face are $\varphi \approx 62°$ for the $Himd^+$ cations in layer A and $\theta \approx 60°$ for those in layer B.

Upon heating the crystal, the $Himd^+$ cations that were frozen at 123 K gradually thaw, as indicated by the large thermal ellipsoids of the cations in the high-temperature range (Supplementary Fig. 3). As revealed by variable-temperature SCXRD measurements, the wobbling of the cation around the molecular $C_5$-axis is accompanied with a reorientation of the molecule. As shown in Fig. 2a, both the angles $\varphi$ and $\theta$, which increase slightly from 123 to 293 K, demonstrate a dramatic increase upon further heating, and finally reach 90° at 393 K. The motion of $Himd^+$ cations actuates significant variations in the cell angles. As shown in Supplementary Fig. 4, the cell angles gradually approach 90°. Correspondingly, the space group of 1 changes to a high-symmetry orthorhombic space group $Pcma$ at 393 K. In the high-temperature phase (HTP), the asymmetric unit contains half a $CuCl_4^{2-}$ dianion and two half $Himd^+$ cations. Notably, the symmetry requirement of $Pcma$ implies that the $Himd^+$ cations in both layers A and B should be rotationally disordered at the HTP. The thermal responses of $Himd^+$ cations in layers A and B are significantly different: the dihedral angle $\theta$ reaches 90° at 373 K, whereas the angle $\varphi$ is only 74° at this temperature (Fig. 2a). Correspondingly, an intermediate-temperature phase (ITP), which belongs to a monoclinic space group $P2_1/c$, is identified in a narrow temperature range (Supplementary Fig. 4, Supplementary Table 1). In the ITP, the asymmetric unit comprises two $Himd^+$ cations and one $CuCl_4^{2-}$ dianion. The symmetry requirement of space group $P2_1/c$ suggests that the $Himd^+$ cations in layer B are rotated in a disordered manner at this temperature.

**Characterization of the phase transition.** The two-step structural phase transitions were further verified by differential scanning calorimetry (DSC) and variable-temperature powder X-ray diffraction (PXRD) studies. The two anomalous peaks present in the heating DSC curve confirm that 1 underwent a two-step phase transition, first at 360 K and then at 387 K. The corresponding exothermic peaks observed at 357 and 384 K in the cooling curve indicate that these phase transitions are reversible (Supplementary Fig. 5). In the variable-temperature PXRD patterns during heating, the peaks at 19.0° and 20.0° at 303 K gradually approach each other and finally merge to become a single peak at 19.2° at 363 K, indicating that the first step of the phase

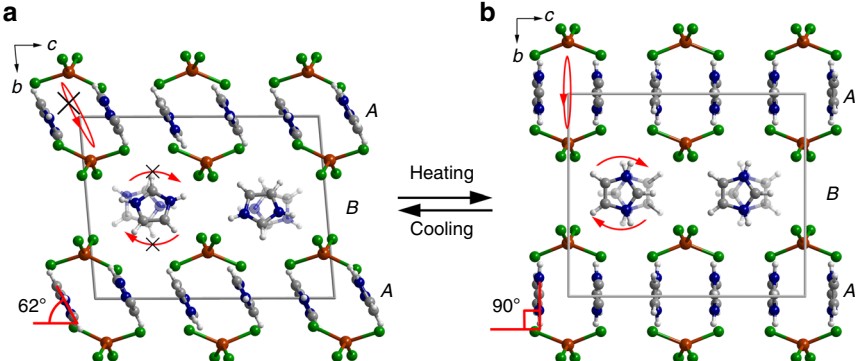

**Fig. 1** Crystal structures of **1** at different temperatures. The basic unit of **1** comprises two imidazole (Himd$^+$) cations and one discrete CuCl$_4^{2-}$ dianion that possesses a heavily distorted tetrahedral geometry. **a** The crystal structure of **1** at 123 K. At this temperature, the Himd$^+$ cation is frozen in a ordered state with a dihedral angle $\varphi$ of 62° relative to the crystallographic (010) face. **b** The crystal structure of **1** at 393 K. In the HTP, the Himd$^+$ cations become rotationally disordered in the molecular plane and the dihedral angle $\varphi$ shifts to 90°. The Himd$^+$ cations in layers A and B are approximately parallel to the crystallographic (001) and (100) faces, respectively. Cu, brown; Cl, green; N, blue; C, gray; H, white-gray

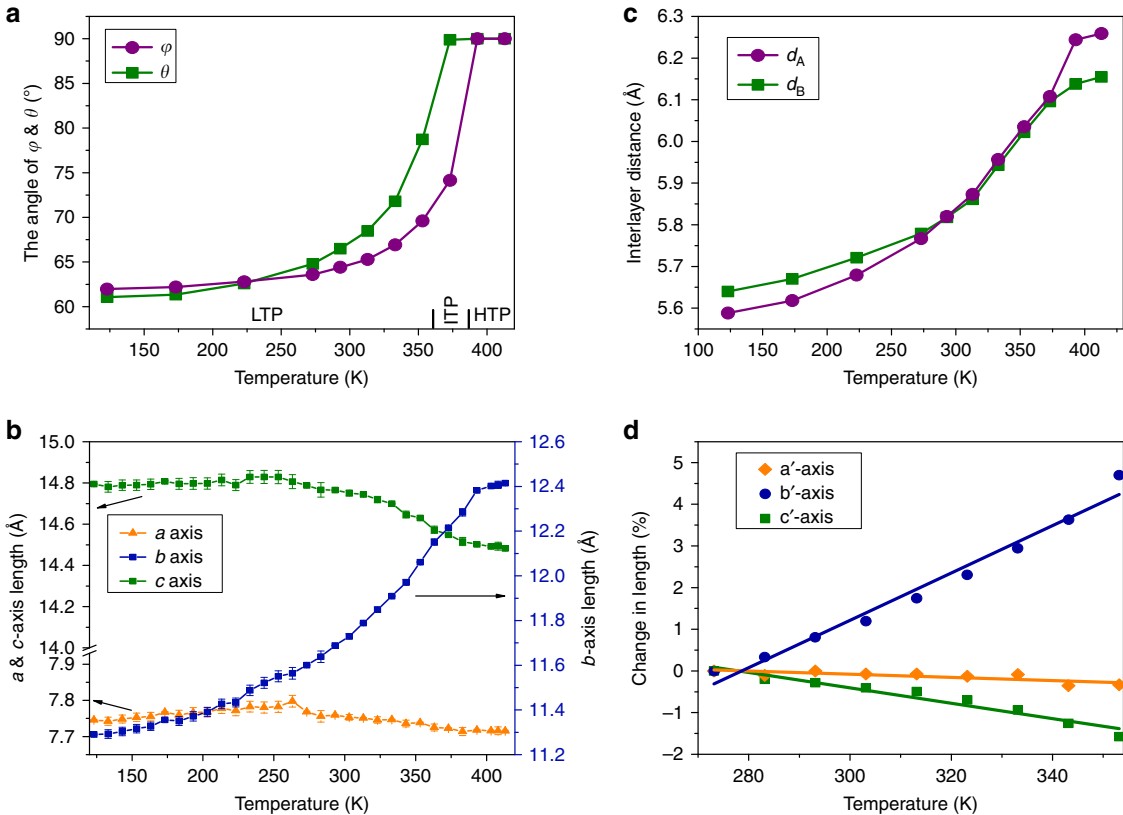

**Fig. 2** Temperature-dependent unit-cell parameters of **1**. **a** The dihedral angles between the Himd$^+$ molecular planes and the crystallographic (010) face. As the temperature increases, both $\varphi$ and $\theta$ gradually increase and finally reach 90° at 393 K. **b** The interlayer distances between CuCl$_4^{2-}$ dianions separated by the Himd$^+$ cations in layers A ($d_A$) and B ($d_B$). **c** The unit-cell length of **1**. Upon heating, the $b$-axis length increases monotonically in the temperature range of 123 to 393 K, whereas the $a$-axis and $c$-axis lengths increase slightly from 123 to 263 K, and then decrease significantly from 273 to 393 K. **d** The anisotropic thermal expansion of the material along the principal axes ($a'$, $b'$, and $c'$) obtained from *PASCal* in the temperature range from 273 to 353 K

transition occurs (Supplementary Figs 6 and 7). Upon further heating, the peaks at 19.2° and 19.6° at 363 K gradually merge together, corresponding to the second step of the phase transition. The continuous shift of the peak position with temperature variation is typical character of second-order phase transitions[27].

**Giant thermal expansion of 1.** The distinct reorientation of circular molecules actuated a significant change in the interlayer distance between the CuCl$_4^{2-}$ dianions separated by the Himd$^+$ cations. As shown in Figs. 2b and 3a, b, the interlayer distances of dianions separated by the cations in A layers ($d_A$) and B layers

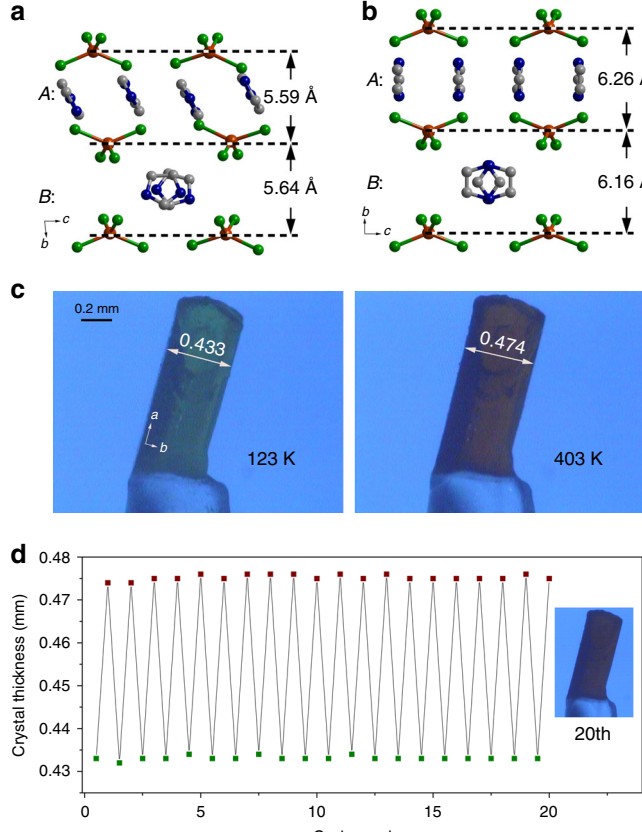

**Fig. 3** The variation of interlayer distance and corresponding crystal deformation. **a** The interlayer distance at 123 K. **b** The interlayer distance at 393 K. Upon heating from 123 to 393 K, the interlayer distance between $CuCl_4^{2-}$ dianions separated by the Himd+ cations in layer A increases from 5.59 to 6.26 Å, while that separated by layer B increases from 5.64 to 6.16 Å. **c** Photographs of the crystal deformation recorded at 123 K (left) and 403 K (right) upon heating. (The length of scale bar is 0.2 mm.) The thickness of the crystal corresponds to the crystallographic *b*-axis. **d** Durability test of the crystal transformation. The inset shows a photograph of the crystal after 20 cycles. The expansion and shrinkage of the crystal thickness were recorded over 20 cycles without significant deterioration in the single-crystal quality

($d_B$) increase from 5.59 to 6.26 Å and 5.64 to 6.16 Å, respectively, upon heating the sample from 123 to 393 K. The increases of $d_A$ and $d_B$ lead to a remarkable increase in the length of the crystallographic *b*-axis. As plotted in Fig. 2c, the *b*-axis elongates by ca. 10% in the temperature range from 123 to 393 K. It is worth mentioning that the change of the *b*-axis length in the range above room temperature is more dramatic than that below room temperature because the reorientation of the Himd+ cation occurs mainly above room temperature. The liner thermal expansion coefficients along the principal axes of material were calculated using the program *PASCal* because of the non-orthogonal crystal lattice in the LTP[28]. As shown in Fig. 2d and Supplementary Fig. 8, a colossal anisotropic thermal expansion with coefficient values of $\alpha_{a'} = -38 \times 10^{-6}$ K$^{-1}$, $\alpha_{b'} = 568 \times 10^{-6}$ K$^{-1}$, and $\alpha_{c'} = -184 \times 10^{-6}$ K$^{-1}$ was observed in the temperature range from 273 to 353 K. The giant PTE along the principal *b*'-axis and NTE along the principal *c*'-axis are among the largest values observed for the solid-state materials (see Supplementary Table 2). Several porous compounds that demonstrate comparable thermal expansion coefficients have been reported; however, an elaborate description of the actuating mechanism at the

molecular level is precluded due to the structural ambiguity of the constituent in the pores[17,18,29]. Due to the colossal PTE along the principal *b*'-axis, a large volumetric thermal expansion with a coefficient of $346 \times 10^{-6}$ K$^{-1}$ was found in the temperature range from 273 to 353 K (Supplementary Fig. 9). The reorientation of molecular cations also induces a sliding movement of anionic layers in the crystallographic (010) plane and a slight change in the molecular structure of $CuCl_4^{2-}$ dianions (Supplementary Figs. 10 and 11). The remarkable sliding movement of molecular layers accounts for the shift of cell angles $\alpha$ and $\gamma$ from 85° to 86° at 123 K to 90° at 393 K.

The significant variation of the cell length implies a distinct transformation of the crystal shape in response to the temperature change. Hence, the deformation of the crystal was examined with a microscope under a continuous $N_2$ gas flow to control the temperature. As shown in Fig. 3c and Supplementary Movie 1, the thickness of the crystal, which corresponds to the crystallographic *b*-axis, expanded 9.5% upon heating from 123 to 403 K, and shrank to the original shape upon cooling to 123 K. The thermally driven reversible shape change was repeated more than 20 times without significant deterioration in the crystal quality, suggesting the remarkable fatigue resistance of this material (Fig. 3d, Supplementary Fig. 12). Notably, the ca. 10% shape transformation demonstrated here is one of the largest shape changes with significant durability observed for solid molecular materials[4,12,18,19,30,31].

**Solid-state nuclear magnetic resonance**. Solid-state $^2$H nuclear magnetic resonance (NMR) spectroscopy is especially effective for the analysis of molecular motion in a wide time scale in solid materials. We therefore performed variable-temperature solid-state $^2$H NMR measurements on a partially deuterated sample of bis($D_3$-imidazolium) tetrachlorocuprate (**2**) to investigate the atypical motion of Himd+ cations. As shown in Fig. 4, the static $^2$H NMR patterns recorded at different temperatures displayed dramatic variations. The simulation of the spectra reveals that the Himd+ cations undergo substantial in-plane wobbling around the molecular $C_5$-axis in the LTP from 250 to 350 K. These patterns can be deconvoluted into two components of in-plane wobbling motions with different angular distributions and amplitudes ($\mu$). The intensity ratio of 6:4 is consistent with the relative populations of Himd+ situated in layers A and B, respectively. As the temperature increases from 250 to 340 K, the wobbling amplitude of the molecular cations gradually increases from 41 ± 3° and 62 ± 7° at 250 K to 65 ± 2° and 88 ± 1° at 340 K (Fig. 4, Supplementary Fig. 13). The intensified in-plane wobbling upon heating is consistent with the large thermal ellipsoids of the cations observed in the SCXRD analyses in the high-temperature range (Supplementary Fig. 3). In the temperature range from 340 to 360 K, apart from the in-plane wobbling motion, a significant out-of-plane vibration of the Himd+ cation ($\sigma$) was identified in the resonance spectra. Such out-of-plane vibration was also detected in the single-crystal structural analyses above 353 K, manifesting as a large out-of-plane component of the thermal ellipsoids in the high-temperature range. At 360 K, the molecules in layer B began full rotation, as indicated by the sharp horns with a quadrupolar splitting of ca. 63 kHz (Fig. 4, Supplementary Fig. 14). In the HTP above 380 K, the molecules in both layers undergo full rotation at a rate faster than $10^7$ Hz. The increasing motional amplitude and eventual full rotation of the Himd+ cation implies an entropy gain of crystal upon heating.

**Discussion**

To elucidate the microscopic origin of the structural variation, we performed Hirshfeld surface analyses to examine the

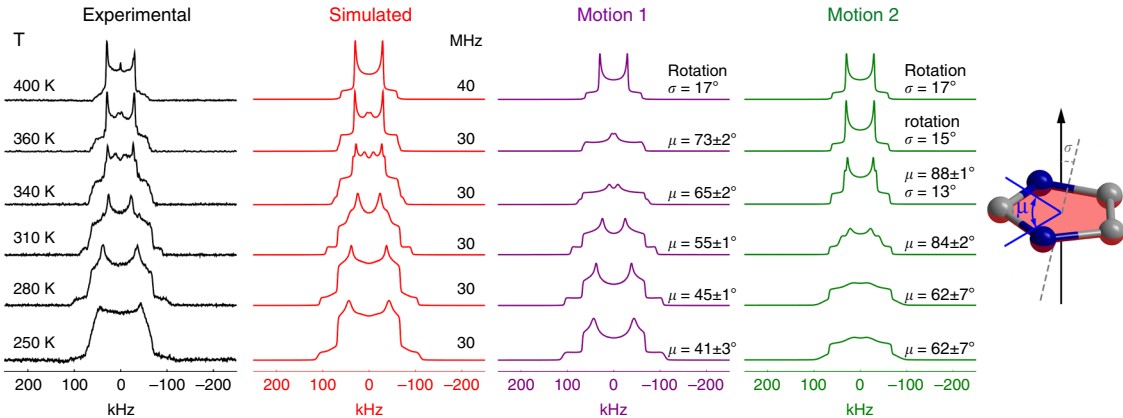

**Fig. 4** $^2$H solid-state NMR spectra of deuterated Himd$^+$ cations at various temperatures. The black and red lines represent the experimental spectra and the simulation results, respectively. According to the simulation results, the $^2$H pattern at each temperature can be deconvoluted into two components with an intensity ratio of 6:4. The motions 1 (purple line) and 2 (green line) should correspond to the different dynamic modes of the Himd$^+$ cations in layers A and B, respectively. The molecular structure depicts the thermodynamic motion of Himd$^+$, where $\mu$ is the mean amplitude of the in-plane wobbling angle around molecular $C_5$-axis and $\sigma$ is the out-of-plane vibration angle

intermolecular interactions in the crystal[32]. As shown in Fig. 5a, the N/C–H⋯Cl hydrogen bonds, which are the most dominant intermolecular interactions around the Himd$^+$ cations, vary significantly as the temperature is increased, with the nearest hydrogen-bond $D$⋯$A$ distance shifting from ca. 3.2 Å in the LTP to ca. 3.5 Å in the HTP. The weaker molecular interactions in the HTP imply a small energy barrier for the rotation of the Himd$^+$ cation in the molecular plane. The 2D fingerprint plots deduced from the Hirshfeld surface reveal that the Himd$^+$ cations and CuCl$_4^{2-}$ complex anions are loosely packed in the crystal (Supplementary Fig. 15). Correspondingly, substantial voids are detected in the crystal structure (Fig. 5b). The calculated volume of the voids increases from 123.03 Å$^3$ at 123 K to 153.06 Å$^3$ at 393 K, implying that the cation can perform various molecular motions in the HTP[33]. Both the weaker hydrogen-bond interactions and the larger free volume around the cation at high temperature are beneficial for the reorientation of Himd$^+$ cations accompanied by molecular rotation. Hence, an entropy-driven phase transition occurs upon heating, with the excess enthalpy in the HTP compensated by the entropy gain from intensified molecular motions.

To support the experimental results, density functional theory (DFT) calculations and classical molecular dynamics (MD) simulations were carried out on **1** to investigate the mechanism of molecular motions. The calculations revealed that the energy of the HTP is higher than that of the LTP by ca. 5.3 kcal/mol per (Himd)$_2$[CuCl$_4$] complex. The larger energy of the HTP is consistent with the above inference that entropy-driven phase transitions occur upon heating, and suggests intermolecular interactions were remarkably reduced in the HTP. Furthermore, the potential energy curve for the rotation of the Himd$^+$ cation along the pseudomolecular $C_5$-axis varies significantly with temperature. As shown in Supplementary Fig. 16, the potential energy barrier for cation rotation decreases from ca. 7.0 kcal/mol at the LTP to 3.5 kcal/mol at the HTP. The smaller energy barrier is possibly overcome by the energetic wobbling motion of the Himd$^+$ cation. As revealed by further MD simulations, the Himd$^+$ cations, which undergo in-plane wobbling in the LTP, rotate dramatically along their pseudo $C_5$-axis in the HTP (Supplementary Fig. 17, Supplementary Movies 2 and 3), agreeing well with the results of variable-temperature SCXRD and solid-state $^2$H NMR measurements. Moreover, the DFT calculations suggest that there are strong electrostatic repulsions between neighboring

Himd$^+$ cations (69.6 kcal/mol for two isolated neighboring Himd$^+$ cations in layer A). As shown in Fig. 5c and Supplementary Fig. 18, the electrostatic repulsions unevenly work on molecular planes of cations when tilted and stacked in an offset manner. Such unbalanced electrostatic repulsions probably drive the reorientation of molecular cations when the intermolecular interactions become weaker in the high temperature.

Many molecule-based materials with thermally controllable molecular rotation have been developed[22–25,34–36]; however, converting the sub-nanoscale molecular motion into a distinct macroscopic mechanical response in the bulk material remains a challenge. In the present material, the circular Himd$^+$ cations act as an entropy-reservoir that change from a statically ordered state into a rotationally disordered state upon heating. The thermodynamic motion of the molecular cations shifts the relative positions of the cations and anions that manifests as collective reorientations of Himd$^+$. Correspondingly, a giant linear PTE along the principal $b'$-axis and NTE along the principal $c'$-axis were detected. Due to the intermolecular interactions between the cations and anions, the molecular level motion propagates into a remarkably macroscopic mechanical response in the bulk material. It should be noted that although the motion of Himd$^+$ cations phenomenologically plays an actuating role, the structural transformation of the material is dominated by synergetic motion of both the cations and anions.

The large thermal expansion actuated by thermodynamically assisted molecular reorientation can be also observed in another molecular crystal, i.e., bis(pyridinium) tetrachlorocuprate (**4**). As shown in Supplementary Fig. 19, the dihedral angles between the molecular planes and crystallographic (010) face slightly increase from 74° at 123 K to 77° at 340 K upon heating and then abruptly shift to 90° at 370 K. The reorientation of pyridinium cations actuates a ca. 7% expansion of crystal along one of its principal axis, most of which occurs in the vicinity of phase transition temperature (4.5%). During the phase transition, the symmetry of crystal **4** changes from a monoclinic space group $C2/c$ to an orthorhombic space group $Cccm$. Both the symmetry variation and large thermal ellipsoids of the cations suggest that the pyridinium cations should be rotationally disordered at a high-temperature phase. Notably, a similar phenomenon cannot be obtained by substituting the Cu(II) metal centers in complex **1** with other transition metals such as Co(II) (Sample **5**) and Zn(II). Hence, the flattening of the tetrahedral geometry of

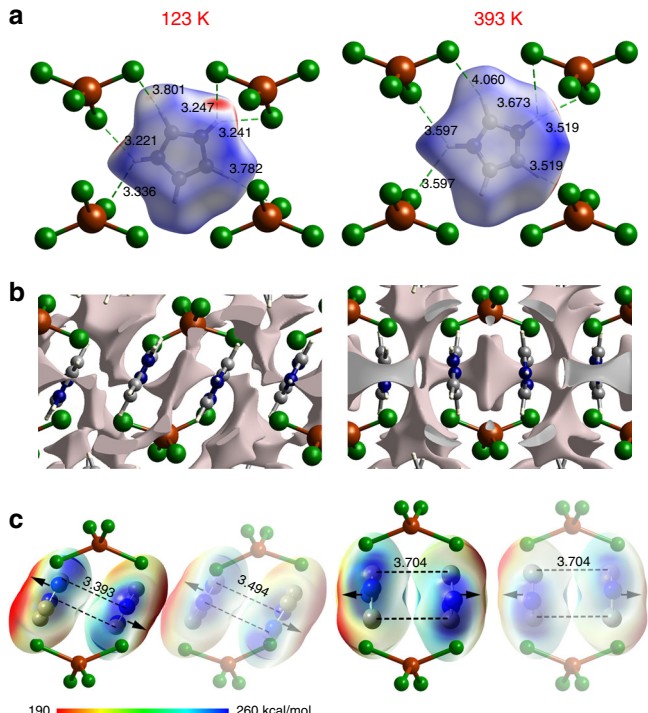

**Fig. 5** Comparison of the molecular interactions in the LTP (left) and HTP (right). **a** The Hirshfeld surface analyses of the Himd$^+$ cation in layer A. The red regions of the surface represent molecular contact shorter than the van der Waals distance. The green dotted lines indicate the potential hydrogen-bond interactions involving Himd$^+$ cations. **b** Void surfaces for the Himd$^+$ cations in layer A. The large voids around the molecular cation allow it to undergo various types of molecular motion. **c** The electrostatic repulsions between the dimer structure of molecular cations in layer A. The electrostatic potential distribution of two cations of the dimer in layer A are shown with isovalue 0.004 a.u. The black arrows denote the dominating electrostatic repulsions on the molecular cations. Notably, the electrostatic repulsions unevenly work on the molecular planes of cations when the cations are in an offset stacking. The molecular cations in different crystallographic planes are drawn in transparency

CuCl$_4^{2-}$ anions due to the Jahn–Teller effect in Cu(II) metal should be essential for the atypical motion of circular molecular cations in these compounds (Supplementary Fig. 20).

In conclusion, we discovered a single-crystal Cu(II) complex possessing a remarkable structural flexibility that manifests as a colossal anisotropic thermal expansion. The large structural transformation is actuated by an atypical reorientation of molecular cations, i.e., the increase in thermal motion of the Himd$^+$ cations on heating appears to cause a change in the relative positions of the cations and anions with consequent change in the orientation of the plane of the Himd$^+$. The result is that the crystal length increases in principal $b'$-axis and decreases in principal $a'$- and $c'$-axes. Such thermodynamically assisted reorientation of circular molecular cations was unambiguously elucidated by variable-temperature SCXRD and solid-state $^2$H NMR measurements. Due to the intermolecular interactions between the cations and anions, the molecular level motion was synchronously transmitted to yield an exceptional large shape change in the bulk material. The ca. 10% expansion and shrinkage of material upon heating and cooling can be reproduced many times without significant deterioration, demonstrating the outstanding fatigue resistance of the material. The exceptional large shape change actuated by the thermodynamically assisted reorientation of circular molecules not only makes the present material a strong candidate for technological applications in future thermal actuators but also will inspire the development of flexible molecular materials or devices with unanticipated physical properties by harnessing molecular thermodynamics.

## Methods

**Bis(imidazolium) tetrachlorocuprate (1).** The yellow single crystals of **1** with large size were grown via slow evaporation of a methanol solution containing CuCl$_2$·2H$_2$O and imidazole hydrochloride in a molar ratio of 1:2 at constant room temperature or diffusion of ethyl ether vapor into the methanol solution[26]. Yield ~80% on the basis of CuCl$_2$·2H$_2$O. Anal. C$_6$H$_{10}$Cl$_4$CuN$_4$ (343.5) Calcd C: 20.98, H: 2.93, N 16.31; found C: 21.11, H: 2.93, N: 16.28.

**Bis(imidazolium-$d_3$) tetrachlorocuprate (2).** The deuterated analog was obtained by slow diffusion of the ethyl ether vapor into a methanol solution containing CuCl$_2$·2H$_2$O and imidazole-$d_3$ hydrochloride with a stoichiometric ratio. The imidazole-$d_3$ hydrochloride was prepared by evaporation of methanol solution containing imidazole-$d_4$ (Sigma-Aldrich, 98 atom% D) and hydrochloric acid under an N$_2$ atmosphere. Yield ~80% on the basis of CuCl$_2$·2H$_2$O. The crystallographic data of this compound are listed in Supplementary Table 3. Anal. C$_6$H$_4$D$_6$Cl$_4$CuN$_4$ (349.5) Calcd C: 20.61, N: 16.03; found C: 20.57, N: 16.11.

**Bis(imidazolium-$d_5$) tetrachlorocuprate (3).** The full deuterated compound **3** was synthesized via the recrystallization of compound **2** twice from methanol-OD solvent under an N$_2$ atmosphere. Result of deuteration was confirmed by IR spectra and temperature-dependent cell parameters of crystal (Supplementary Figs. 21 and 22). (The crystallographic data are listed in Supplementary Table 4.) Anal. C$_6$D$_{10}$Cl$_4$CuN$_4$ (353.5) Calcd C: 20.38, N: 15.84; found C: 20.47, N: 16.10.

**Compounds 4 and 5.** The samples of bis(pyridinium) tetrachlorocuprate (**4**) and bis(imidazolium) tetrachlorocobaltate (**5**) were synthesized with the same method as that for the sample **1** was used. Yield ~70% and 75% on the basis of metal reactants, respectively. The crystallographic data of compounds **4** and **5** are listed in Supplementary Tables 5 and 6, respectively. Anal. (**4**) C$_{10}$H$_{12}$Cl$_4$CuN$_2$ (365.6) Calcd C: 32.85, H: 3.31, N 7.66; found C: 32.57, H: 3.55, N: 7.57. (**5**) C$_6$H$_{10}$Cl$_4$CoN$_4$ (338.9) Calcd C: 21.26, H: 2.97, N 16.53 found C: 21.05, H: 2.93, N: 16.29.

**Solid-state NMR method.** The $^2$H NMR experiments were performed on a Bruker 400WB AVANCE spectrometer at the field of 9.4 T. The 2H signals were acquired at the Larmor frequency of 61.42 MHz using a 3.2 mm triple resonance MAS probe. Quadrupolar echo sequence 90°$_x$–$\tau_1$–90°$_y$–$\tau_2$-acquisition was employed to get static $^2$H NMR spectrum. The 90° pulse length of 2.6 μs was sufficiently short to allow spectral acquisition with minimal spectral distortion. $\tau_1$ was set to 100–200 μs for the experiment of different temperatures. In all, 1000 to 3000 scans were accumulated for each spectrum depending on the signal-to-noise-ratio. The temperature was controlled by a Bruker temperature controller BCU II with a deviation of ±1.0 K. Simulations were carried out using the EXPRESS package in MATLAB[37].

**Single-crystal X-ray diffraction.** SCXRD analyses of **1** were performed on a Rigaku-CCD diffractometer equipped with a Mo-Kα radiation source. The sample was cooled or warmed under a continuous flow of cold or hot N$_2$ gas generated using a Rigaku GN$_2$ low-temperature apparatus. The data from one sample were collected at 123, 173, 223, 273, 293, 313, 333, 353, 373, 393 and 413 K, in sequence. The diffraction data of samples **2**, **3**, **4** and **5** were collected on a Rigaku XtaLab Pro. The structures were solved by a direct method and refined by full-matrix least-squares on $F^2$ using the SHELX program[38] with anisotropic thermal parameters for all non-hydrogen atoms. The hydrogen atoms were geometrically added and refined by the riding model. The non-hydrogen atoms in the imidazolium cation were refined with 60% C and 40% N when they are rotated in the ITP and HTP.

**Differential scanning calorimetry.** DSC measurements were performed on PerkinElmer DSC 8000 instrument using cooling and heating rates of 30 K/min.

**Calculation details.** The classical MD simulations were performed by the *Forcite* program implemented in the *Materials Studio 5.5* package[39]. The constant-volume & temperature (NVT) ensemble was performed to simulate the dynamic processes at selected temperature. The partial atomic charges of all the atoms were estimated through the charge equilibration (QEq) method, and the Universal force field with Nose thermostat and random initial velocities method were used. The electrostatic interactions and the van der Waals interactions were evaluated by the Ewald summation method, with a Buffer width of 0.5 Å. The total simulation time was 2 ns, with a time step of 1.0 fs. The rotation energies were calculated by the periodic DFT using the 6.1 version of Quantum ESPRESSO[40,41]. The widely generalized gradient approximation of the Perdew–Burke–Ernzerhof functional[42] and the Blöchl all-electron projector augmented wave method[43] were employed. Plane

wave basis sets with a cutoff energy of 500 eV were used for all calculations. Brillouin zone sampling was restricted to the Γ point. The BFGS quasi-Newton algorithm method based on the trust radius procedure was used for geometry optimizations. The climbing-image nudged elastic band method[44] with the quasi-Newton Broyden's second algorithm was used to determine a minimum energy path and to locate a first-order saddle point that corresponds to a transition state. The DFT including the long-range dispersion correction (DFT-D) was also taken into account using the Grimme semiempirical method[45] to describe the long-range van der Waals interactions. For all the DFT-D calculations, the energy and force convergence criterions were set as $1 \times 10^{-4}$ Ry and $1 \times 10^{-3}$ Ry/Bohr, respectively. The tentative energy decomposition of the isolated imidazolium pair was performed at B3LYP/6–311G* level[46,47] as implemented in the Gaussian 09 suite[48]. The positions of hydrogen atoms were optimized while all the other coordinates were frozen. The orbital interaction energy term was subtracted from the total energy change when forming the pair, yielding the electrostatic and exchange repulsion of 69.6 kcal/mol for the pair in layer A at 123 K and 71.6 kcal/mol at 398 K. The electrostatic potential calculation were performed at ω-B97XD/6–31G* level[49].

## Data availability

All data generated and analyzed in this study are included in the article and its Supplementary Information, and are also available from the authors upon request. The X-ray crystallographic coordinates for structures reported in this study have been deposited at the Cambridge Crystallographic Data Centre (CCDC) under deposition numbers CCDC1870817 (**1** at 123 K, LTP), 1870557 (**1** at 173 K, LTP), 1870563 (**1** at 223 K, LTP), 1870556 (**1** at 273 K, LTP), 1870555 (**1** at 293 K, LTP), 1870559 (**1** at 313 K, LTP), 1870564 (**1** at 333 K, LTP), 1870562 (**1** at 353 K, LTP), 1870558 (**1** at 373 K, ITP), 1870560 (**1** at 393 K, HTP) and 1870561 (**1** at 413 K, HTP); 1870569 (**2** at 123 K, LTP), 1870568 (**2** at 293 K, LTP), 1870566 (**2** at 360 K, ITP) and 1870567 (**2** at 395 K, HTP); 1915206 (**3** at 123 K, LTP), 1915206 (**3** at 293 K, LTP), 1915204 (**3** at 373 K, ITP) and 1915205 (**3** at 393 K, HTP); 1870571 (**4** at 123 K, LTP), 1870570 (**4** at 340 K, ITP) and 1870572 (**4** at 370 K, HTP); 1870575 (**5** at 123 K, LTP) and 1870574 (**5** at 390 K, HTP). These data can be obtained free of charge via http://www.ccdc.cam.ac.uk/data_request/cif.

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

## Acknowledgements

This work was supported by the National Natural Science Foundation of China (Grants 21701013, 21325103 and 21671161), MEXT KAKENHI (Grant Number JP17H01197), and Beijing Institute of Technology Research Fund Program for Young Scholars.

## Author contribution

Z.-S.Y., O.S. and J.T. designed the study, conducted experiments and wrote most of the paper. H.G. and X.K. performed the solid-state NMR measurements and wrote the related discussion. Y.S., K.Y., C.-T.H., S.-Q.W. and X.-Y.Z. performed the calculations. X.-L.W. and S.-Q.S. assisted in structural measurements and results analyses. All authors discussed the results and commented on the manuscript.

## Competing interests

The authors declare no competing interests.
