## [Peer Review File · Nature Communications]

Reviewers' comments:

Reviewer #1 (Remarks to the Author):

The authors describe extremely large anisotropic thermal expansion in a simple crystalline material. Without summarising all their findings here, I believe that the results are interesting and well presented. The manuscript is very well written and the authors have carried out a very detailed analysis of their crystal structures and data from supporting analyses. There is no doubt that the presentation style is suitable for Nature Communications and that the results will be deemed interesting and important by the community.

My main concern is that the material undergoes two phase transitions in the temperature range for which the thermal expansion parameters are reported. Other studies in the field calculate the coefficients for a single phase rather than over different phases. If the community were to calculate thermal expansion parameters by including phase changes then the phenomenon of "anomalous" thermal expansion would become rather unremarkable because there are many materials that would appear to have similarly "remarkable" properties.

My other concern is that the authors imply in the beginning of the manuscript that the material was designed, and then later that it was discovered. Which is it?

Until the authors have addresses the above mentioned concerns, I do not support publication of this work.

Reviewer #2 (Remarks to the Author):

The article reports significant, simultaneous, directionally-dependent negative and positive thermal expansion in crystals of $(\text{Hmd})_2[\text{CuCl}_4]$.

The authors report a detailed temperature dependent single crystal X-ray study and use this to show the proposed mechanism of the thermal expansion, accounting for the very large simultaneous positive and negative thermal expansion. The proposed mechanism is supported by solid state NMR

and DFT calculations. The experimental work appears to have been carried out thoroughly and with significant attention to detail.

As a relatively simple salt lacking the 3D connectivity of MOFs and other materials known to display such significant thermal expansion characteristics I feel that this work is relatively novel and will be of wide interest in materials chemistry.

I find the work and the compound to be compelling and worthy of publication in Nature Communications but not in this manuscript as it currently reads.

I have a couple of issues with the conclusions from the paper, namely the proposed mechanism, and I think that there needs to be some clarity/change in the manuscript in this context. I don't think the current mechanism is sound.

As I understand it, the authors are claiming that on heating, the Hmid cations begin to rotate and as a result, they "stand up", similar (but in reverse) to the way a rolling bicycle wheel resists toppling. As a consequence the molecules change their orientation in space and extend the dimensions of the crystal (and hence unit cell) in one direction while substantially reducing the dimension of the crystal in orthogonal direction. The changes in molecular orientation are clearly responsible for the expansion characteristics of the crystal but I am not convinced about the cause of this.

I am recalling 1st year physics (from a long time ago and possibly not very well) but it seems to me that rotation of molecules would cause a gyroscopic effect and they would actually resist change in orientation. I don't think gravitational effects will be significant enough at the molecular level to influence a change in the plane of rotation (as happens with the rolling wheel analogy). In any case - the crystals are not oriented specifically to align this gravitational influence.

Furthermore - there is no reason for the molecules to rotate principally in one direction - any motion would be Brownian (strictly applies to fluids but the molecules would act similarly in the crystal void but with restricted translational motion) and therefore random - possibly rotating backwards and forwards randomly rather than rotating continuously and uni-directionally around an axis as would be needed to maintain net angular momentum.

It should be enough to describe the rearrangements and the subsequent changes of phase as responsible for the large thermal expansion without trying to convolute the mechanism with potentially wrong/misleading analogies.

I think the compound and the extent of the thermal expansion warrants publishing in this journal but not with the current proposed mechanism. It is possible that I have misunderstood the proposed mechanism. I will be interested to see the comments of other referees in this regard and remain open to being convinced of the veracity of the mechanism but at this point in time I cannot recommend publication in the current form.

Another small point of contention is the claim that the "Himd should be rotated in the HTP to satisfy the requirement of the space group". This is wrong in terms of cause and effect. The space group is a consequence of the position of the molecules. Molecules do not organise to meet requirements of space groups (which is just a mathematical construct for simplifying the description of the relative arrangements of objects in space).

Reviewer #3 (Remarks to the Author):

The work by Yao et al describes a thermoresponsive single crystal material with highly reversible anisotropic expansion. The authors posit that an "entropy reservoir" of sorts allows increasing thermal motion at high temperatures, which causes changes in the crystalline lattice. In particular, they extensively study and characterize the dynamics of an imidazole cation intercalated between rows of copper chlorides.

The authors do a very good job at characterizing their system. The results are interesting and the area of thermosalient systems is a broadly interesting and developing field, for which this journal would make a good home; however, in some cases, the authors make claims that are unsubstantiated. Prior to publication, these statements should be reconsidered by the authors.

We focused on circular molecules as an entropy-reservoir

60 unit; if the constituent circular molecules stand up when rotated and topple over when

61 the rotation is ceased, just like the precession-related motion of a rolling hoop

62 (Scheme S1), such molecular motion potentially induces a significant variation in the

63 lattice length.

The authors appear to be making a claim that gyroscopic forces are causing the molecules into an "upright" position, which seems implausible. Further, the entirety of their paper focuses on rocking motions of these molecules, not rotating motions. This text and Scheme S1 should be removed or greatly clarified so as to avoid confusion.

Further:

Hence, an

224 entropy-driven phase transition occurs upon heating, with the excess enthalpy in the

225 HTP compensated by the entropy gain from intensified molecular motions.

This statement could be supported better. While the behavior of the imidazole is well characterized, the imidazole motion is likely not the only dynamics in the crystal. Further comment and study are needed. As the authors are aware, phase transitions are a cumulative effect of the entire solid state structure, the authors should further their due diligence into how the energetics of reorienting the copper chlorides are affected by the more parallel orientation of the imidazole. Might this reorientation that would make the overall process enthalpic? I do not know and feel as though the best we can do with the data presented by the authors is speculate.

245 Preliminary experiments reveal that the large anisotropic thermal expansion

246 induced by the reorientation of molecular wheels can also be realized if the Himd+

247 cation is replaced with protonated pyridine (Sample 3, see Supplementary Fig. 17).

Preliminary experiments are great for grants, but I would avoid them in papers. The work should be completed and fully characterized. This is an important aspect of the generalizability of their claims and deserves more than a brief and passing statement.

In the SI:

The deuteration of the sample induces slightly decrease in the phase transition temperatures,

103 suggesting the hydrogen bond interactions in the crystal play minor roles in the phase

104 transitions.

A reference for this statement and further elaboration is needed.

Response to reviewer 1:

Thank you for meticulously reading our manuscript and for providing valuable comments to improve it. We have revised the manuscript according to the reviewers' comments.

Reviewer #1 (Remarks to the Author):

The authors describe extremely large anisotropic thermal expansion in a simple crystalline material. Without summarising all their findings here, I believe that the results are interesting and well presented. The manuscript is very well written and the authors have carried out a very detailed analysis of their crystal structures and data from supporting analyses. There is no doubt that the presentation style is suitable for Nature Communications and that the results will be deemed interesting and important by the community.

My main concern is that the material undergoes two phase transitions in the temperature range for which the thermal expansion parameters are reported. Other studies in the field calculate the coefficients for a single phase rather than over different phases. If the community were to calculate thermal expansion parameters by including phase changes then the phenomenon of "anomalous" thermal expansion would become rather unremarkable because there are many materials that would appear to have similarly "remarkable" properties.

Answer: Thank you for your valuable comments. Indeed, most studies in this field calculated thermal coefficients for a single phase; however, the unusual thermal expansion behavior of these reported materials is often accompanied with a phase transition (Mary T. A. *et al. Science* **1996**, 272, 90-92; Das S. *et al. Nature Mater.* **2009**, 9, 36-39; Panda M. K. *et al.*

Nature Commun. **2014**, 4811). This finding implies that the phase transition play significant roles in the unusual thermal expansion. In these compounds, the thermal coefficients were calculated in a single phase because the discontinuous or abrupt variation in the cell length was often detected at the phase transition temperature. In the present compounds, the large anisotropic thermal expansion can be actuated by thermodynamic rotation-assisted reorientation of molecular wheels. The molecular wheels are reoriented in different phases, and more importantly, the cell lengths gradually increase with no obvious discontinuity at the phase transition point (Figure 2c). Therefore, the thermal expansion coefficients were calculated in different phases to elucidate the large thermal expansion induced by reorientation of molecular wheels. However, the difference in the calculation of thermal expansion coefficient between our case (two phases) and a typical case (single phase) should be indicated. Therefore, we have added the following sentences:

Note that the thermal expansion coefficients were calculated in different phases to elucidate the large thermal expansion induced by the reorientation of molecular wheels. This is different from the typical estimation of thermal expansion coefficients reported previously^{1,2,4-6}, wherein the thermal expansion coefficients were calculated for a single phase.

My other concern is that the authors imply in the beginning of the manuscript that the material was designed, and then later that it was discovered. Which is it?

Until the authors have addresses the above mentioned concerns, I do not support publication of this work.

Answer: Thank you for your insightful comments. Our previous report on a single crystal of Ni(II) complex, $[\text{Ni}^{\text{II}}(\text{en})_3](\text{ox})$ (en, ethylenediamine; ox^{2-} , oxalate anion) (*Nature chem.* **2014**, 6, 1079-1083) motivated us to develop a dynamic single crystal equipped with molecular wheels. In that crystal, ox^{2-} can perform 90° reorientation accompanied by a rotationally disordered–statically ordered structural variation. The rotation-assisted reorientation of ox^{2-} drives an abrupt shape change of single crystal. The thermodynamic rotation is also observed in circular molecules, and we envisaged that such rotation potentially induces the reorientation of the molecular plane, triggering a larger change in the cell length (as illuminated in revised Scheme 1 in supporting information). With this idea in mind, we conducted this study.

To clarify this point, we modified the related content in the introduction according to reviewers' comments.

We have previously discovered that oxalate anions (ox^{2-}) can perform atypical 90° rotation accompanied with a rotationally disordered–statically ordered structural variation in a single crystal of Ni(II) complex¹⁹. The dramatic reorientation of ox^{2-} , which is dominated by competition between molecular thermodynamic rotation and subtle intermolecular interactions, leads to a moderate expansion/contraction (4.9%) in the shape of a single crystal. Thermodynamic rotation is also observed in circular molecules²²⁻²⁵; hence, in this study, we focus on circular molecules as an entropy-reservoir unit and envisage that the orientation of molecular planes can be modulated by a competition between the in-plane thermodynamic rotation and non-coplanar intermolecular interactions (Supplementary Scheme 1). If the constituent circular molecules stand up when rotated and topple over when the rotation is ceased, such a molecular motion may induce a significant variation in the lattice length.

The scheme 1 in SI was also modified to clarify this research strategy:

Supplementary Scheme 1. The rotation assists reorientation in real top and wheel (a) and molecular systems (b). The oxalate anion, which is rotationally standing-up and statically lying down, is reminiscent of spinning tops. Apart from tops, the rotation-assisted reorientation is also observed in rolling hoops; hence, in this study, we were motivated to investigate circular molecules as an entropy-reservoir unit. Notably, in contrast to the unidirectional rotation of top and wheel, the rotation of molecules should be in a Brownian manner, randomly forward and backward.

Response to reviewer 2:

Thank you for meticulously reading our manuscript and for providing valuable comments to improve it. We have revised the manuscript according to the reviewers' comments.

Reviewer #2 (Remarks to the Author):

The article reports significant, simultaneous, directionally-dependent negative and positive thermal expansion in crystals of $(\text{Himd})_2[\text{CuCl}_4]$. The authors report a detailed temperature dependent single crystal X-ray study and use this to show the proposed mechanism of the thermal expansion, accounting for the very large simultaneous positive and negative thermal expansion. The proposed mechanism is supported by solid state NMR and DFT calculations. The experimental work appears to have been carried out thoroughly and with significant attention to detail.

As a relatively simple salt lacking the 3D connectivity of MOFs and other materials known to display such significant thermal expansion characteristics I feel that this work is relatively novel and will be of wide interest in materials chemistry.

I find the work and the compound to be compelling and worthy of publication in Nature Communications but not in this manuscript as it currently reads.

I have a couple of issues with the conclusions from the paper, namely the proposed mechanism, and I think that there needs to be some clarity/change in the manuscript in this context. I don't think the current mechanism is sound.

As I understand it, the authors are claiming that on heating, the Hmid cations begin to rotate and as a result, they "stand up", similar (but in reverse) to the way a rolling bicycle wheel resists toppling. As a consequence the molecules change their orientation in space and extend the dimensions of the crystal (and hence unit cell) in one direction while substantially reducing the dimension of the crystal in orthogonal direction. The changes in molecular orientation are clearly responsible for the expansion characteristics of the crystal but I am not convinced about the cause of this.

I am recalling 1st year physics (from a long time ago and possibly not very well) but it seems to me that rotation of molecules would cause a gyroscopic effect and they would actually resist change in orientation. I don't think gravitational effects will be significant enough at the molecular level to influence a change in the plane of rotation (as happens with the rolling wheel analogy). In any case - the crystals are not oriented specifically to align this gravitational influence.

Furthermore - there is no reason for the molecules to rotate principally in one direction - any motion would be Brownian (strictly applies to fluids but the molecules would act similarly in the crystal void but with restricted translational motion) and therefore random - possibly rotating backwards and forwards randomly rather than rotating continuously and uni-directionally around an axis as would be needed to maintain net angular momentum.

It should be enough to describe the rearrangements and the subsequent changes of phase as responsible for the large thermal expansion without trying to convolute the mechanism with potentially wrong/misleading analogies.

I think the compound and the extent of the thermal expansion warrants publishing in this journal but not with the current proposed mechanism. It is possible that I have misunderstood the proposed mechanism. I will be interested to see the comments of other referees in this

regard and remain open to being convinced of the veracity of the mechanism but at this point in time I cannot recommend publication in the current form.

Answer: Thanks for your detailed and valuable comments. You are absolutely right; the gravity does not play significant roles during the molecular reorientation, and the molecular wheels should rotate in Brownian style, randomly forward or backward. We apologize for this misleading interpretation.

In our first submitted manuscript, we considered that the rotation of imidazolium around its C_5 axis changes the orientation of the molecular plane. Therefore, this molecular motion was compared with rolling hoop and the word “precession” was used to define this type of molecular motion because precession or gyroscopic effect is defined in the dictionary as “*a change in the orientation of the rotational axis of a rotating body.*” However, as suggested by you and reviewer 3, this definition may confuse the mechanism and lead to a misunderstanding that the gravity significantly acts on the molecular reorientation. Therefore, this discussion was deleted from the revised manuscript and appropriate modifications were made as follows:

We have previously discovered that oxalate anions (ox^{2-}) can perform atypical 90° rotation accompanied with a rotationally disordered–statically ordered structural variation in a single crystal of Ni(II) complex¹⁹. The dramatic reorientation of ox^{2-} , which is dominated by competition between molecular thermodynamic rotation and subtle intermolecular interactions, leads to a moderate expansion/contraction (4.9%) in the shape of a single crystal. Thermodynamic rotation is also observed in circular molecules²²⁻²⁵; hence, in this study, we focus on circular molecules as an entropy-reservoir unit and envisage that the orientation of molecular planes can be modulated by a competition between the in-plane thermodynamic

rotation and non-coplanar intermolecular interactions (Supplementary Scheme 1). If the constituent circular molecules stand up when rotated and topple over when the rotation is ceased, such a molecular motion may induce a significant variation in the lattice length.

The related discussion in the text was also revised to clarify this point. Because the scheme 1 in supporting information may provide some information to help understand our strategy and establish a relation with our previous work (as questioned by reviewer 1), we retained scheme S1 after making some significant improvements.

Supplementary Scheme 1. The rotation assists reorientation in real top and wheel (a) and molecular systems (b). The oxalate anion, which is rotationally standing-up and statically lying down, is reminiscent of spinning tops. Apart from tops, the rotation-assisted reorientation is also observed in rolling hoops; hence, in this study, we were motivated to investigate circular molecules as an entropy-reservoir unit. Notably, in contrast to the unidirectional rotation of top and wheel, the rotation of molecules should be in a Brownian manner, randomly forward and backward.

Another small point of contention is the claim that the "Himd should be rotated in the HTP to satisfy the requirement of the space group". This is wrong in terms of cause and effect. The space group is a consequence of the position of the molecules. Molecules do not organise to meet requirements of space groups (which is just a mathematical construct for simplifying the description of the relative arrangements of objects in space).

Answer: Thank you for your suggestion; this sentence was reorganized to improve the logicity as follows:

The motion of Himd⁺ cations induces significant variations in the cell angles. As shown in Supplementary Fig. 4, the cell angles gradually approach 90°. Correspondingly, the space group of **1** changes to a high-symmetry orthorhombic space group *Pcma* at 393 K. In the high-temperature phase (HTP), the asymmetric unit contains half a CuCl₄²⁻ dianion and two half Himd⁺ cations. Notably, the symmetry requirement of *Pcma* implies that the Himd⁺ cations in both layers A and B should be rotationally disordered at the HTP.

Response to reviewer 3:

Thank you for meticulously reading our manuscript and for providing valuable comments to improve it. We have revised the manuscript according to the reviewers' comments.

Reviewer #3 (Remarks to the Author):

The work by Yao et al describes a thermoresponsive single crystal material with highly reversible anisotropic expansion. The authors posit that an "entropy reservoir" of sorts allows increasing thermal motion at high temperatures, which causes changes in the crystalline lattice. In particular, they extensively study and characterize the dynamics of an imidazole cation intercalated between rows of copper chlorides.

The authors do a very good job at characterizing their system. The results are interesting and the area of thermosalient systems is a broadly interesting and developing field, for which this journal would make a good home; however, in some cases, the authors make claims that are unsubstantiated. Prior to publication, these statements should be reconsidered by the authors.

We focused on circular molecules as an entropy-reservoir

60 unit; if the constituent circular molecules stand up when rotated and topple over when
61 the rotation is ceased, just like the precession-related motion of a rolling hoop
62 (Scheme S1), such molecular motion potentially induces a significant variation in the
63 lattice length.

The authors appear to be making a claim that gyroscopic forces are causing the molecules into an "upright" position, which seems implausible. Further, the entirety of their paper focuses on

rocking motions of these molecules, not rotating motions. This text and Scheme S1 should be removed or greatly clarified so as to avoid confusion.

Answer: Thank you for your positive comments and valuable suggestions. In the first submitted manuscript, we used the term “precession” to describe the thermodynamic rotation-assisted reorientation of molecular wheels because the orientation of molecular wheel can be controlled by a rotation around molecular C_5 axis. However, as suggested by you and reviewer 2, this term is inappropriate and misleading. Therefore, it was deleted in the revised text and related discussion was modified as follows.

We have previously discovered that oxalate anions (ox^{2-}) can perform atypical 90° rotation accompanied with a rotationally disordered–statically ordered structural variation in a single crystal of Ni(II) complex¹⁹. The dramatic reorientation of ox^{2-} , which is dominated by competition between molecular thermodynamic rotation and subtle intermolecular interactions, leads to a moderate expansion/contraction (4.9%) in the shape of a single crystal. Thermodynamic rotation is also observed in circular molecules²²⁻²⁵; hence, in this study, we focus on circular molecules as an entropy-reservoir unit and envisage that the orientation of molecular planes can be modulated by a competition between the in-plane thermodynamic rotation and non-coplanar intermolecular interactions (Supplementary Scheme 1). If the constituent circular molecules stand up when rotated and topple over when the rotation is ceased, such a molecular motion may induce a significant variation in the lattice length.

Because Scheme 1 in supporting information may assist in understanding the origin of this research strategy and connect this study with our previous study (as questioned by reviewer 1), Scheme S1 was retained after making considerable improvements.

Supplementary Scheme 1. The rotation assists reorientation in real top and wheel (a) and molecular systems (b). The oxalate anion, which is rotationally standing-up and statically lying down, is reminiscent of spinning tops. Apart from tops, the rotation-assisted reorientation is also observed in rolling hoops; hence, in this study, we were motivated to investigate circular molecules as an entropy-reservoir unit. Notably, in contrast to the unidirectional rotation of top and wheel, the rotation of molecules should be in a Brownian manner, randomly forward and backward.

Further: Hence, an

224 entropy-driven phase transition occurs upon heating, with the excess enthalpy in the
225 HTP compensated by the entropy gain from intensified molecular motions.

This statement could be supported better. While the behavior of the imidazole is well characterized, the imidazole motion is likely not the only dynamics in the crystal. Further comment and study are needed. As the authors are aware, phase transitions are a cumulative effect of the entire solid state structure, the authors should further their due diligence into how the energetics of reorienting the copper chlorides are affected by the more parallel orientation

of the imidazole. Might this reorientation that would make the overall process enthalpic? I do not know and feel as though the best we can do with the data presented by the authors is speculate.

Answer: Your illuminating suggestion is greatly appreciated. As rightly suggested, the phase transitions are a cumulative effect of the entire solid-state structure; hence, we further examined the structural variation of the whole crystal. As shown in the figure below (Supplementary Fig. 10 in revised SI), the anionic layers undergo a significant displacive motion accompanied with the reorientation of molecular wheels in the direction that is perpendicular to the anionic layer and in a plane that is parallel to the anionic layer (crystallographic *ac* plane). The sliding movement of the layers in the plane accounts for the shift of cell angles α and γ from 85° – 86° at 123 K to 90° at 393 K. The reorientation of molecular wheels induces a change in the orientation of CuCl_4^{2-} dianions. However, the orientational change in dianions is much smaller than that in the molecular wheels with respect to the crystallographic (010) plane. In addition to the orientational change, we examined the molecular structural change of dianions. As shown in Supplementary Fig. 11, both Cu–Cl bond lengths and Cl–Cu–Cl angles demonstrate a small change upon heating from 123 to 393 K, suggesting that the structure of dianions is not considerably affected by crystal transformation. The small changes in orientation and molecular structure of CuCl_4^{2-} dianions suggest that the dianions play major roles as molecular gear racks, transmitting the reorientation of molecular wheels to a large thermal expansion of bulk crystal.

Based on your suggestions, we further performed the DFT calculations to investigate the energy change during crystal structural transformation. The calculation results reveal that the energy difference between the HTP and LTP was $-1.13 \text{ kJ mol}^{-1}$ per $(\text{Himd})_2[\text{CuCl}_4]$ complex. However, the lower energy of HTP is inconsistent with endothermic peaks found at

the phase transition point upon heating in the DSC curve. Indeed, we think the current theoretical calculations cannot provide accurate energy information in such highly dynamical system because (1) the intermolecular interactions, which should be averaged over all the possible configurations as the molecular cations are rotationally disordered at the HTP, cannot be exactly calculated using current theoretical calculation methods and (2) the kinetic energy of the cations under unbounded rotation, which was generally discarded in typical DFT calculations, may significantly contribute to the energy of HTP, particularly for a system with delicate energy change. Although the DFT calculation results suggest that the energy of HTP is slightly lower than that of the LTP, based on the endothermal peaks found in the DSC curves upon heating and principals of general chemistry, *i.e.*, the Gibbs energy that determines the phase transition direction, and the entropy-driven reaction that usually occurs upon heating, we ascribe our observation to an entropy-driven structural phase transition.

Supplementary Fig. 10 | Superimposed drawing of the crystal structures in the HTP (393 K, red) and LTP (123 K, green). The panoramic view of crystal structure reveals that the reorientation of molecular wheels actuates a large expansion in the interlayer distance of CuCl_4^{2-} dianions and induces a significant sliding movement of the layers in the

crystallographic (010) plane. Notably, the orientational change in dianions is much smaller than that in the molecular wheels with respect to the crystallographic (010) plane.

Supplementary Fig. 11 | The molecular structure of CuCl_4^{2-} dianions. The large structural transformation of crystal induces a small change in the Cu-Cl bond lengths and Cl-Cu-Cl angles. Both the small changes in orientation and the molecular structure of CuCl_4^{2-} dianions suggest the dianions play major roles as molecular gear racks, connecting the molecular cations via intermolecular interactions and transmitting the motion of molecular wheels into a giant anisotropic thermal expansion of bulk crystal.

The above figures have been added in the supporting information, and the following sentence has been added in the revised manuscript:

The reorientation of molecular wheels also induces a sliding movement of anionic layers in the crystallographic (010) plane, which accounts for the shift of cell angles α and γ from 85°–86° at 123 K to 90° at 393 K (Supplementary Fig. 10).

245 Preliminary experiments reveal that the large anisotropic thermal expansion
246 induced by the reorientation of molecular wheels can also be realized if the Himd+
247 cation is replaced with protonated pyridine (Sample 3, see Supplementary Fig. 17).

Preliminary experiments are great for grants, but I would avoid them in papers. The work should be completed and fully characterized. This is an important aspect of the generalizability of their claims and deserves more than a brief and passing statement.

Answer: Thank you for your suggestions. The thermal expansion properties of Bis(pyridinium) tetrachlorocuprate was further investigated, and the following related discussion was added in the text:

The large thermal expansion induced by thermodynamically assisted molecular reorientation can be also observed in another molecular crystal, *i.e.*, bis(pyridinium) tetrachlorocuprate (**4**). As shown in Supplementary Fig. 19, the dihedral angles between the molecular planes and crystallographic (010) face slightly increase from 74° at 123 K to 77° at 340 K upon heating and then abruptly shift to 90° at 370 K. The reorientation of pyridinium cations induces a *ca.* 7% expansion of crystal along one of its principal axis, most of which occurs in the vicinity of phase transition temperature (4.5%). During the phase transition, the symmetry of crystal **4** changes from a monoclinic space group $C2/c$ to an orthorhombic space group $Cccm$. Both the symmetry variation and large thermal ellipsoids of the cations suggest that the pyridinium cations should be rotationally disordered at a high-temperature phase.

Supplementary Fig. 19 was also improved as shown below:

Supplementary Fig. 19 | The large anisotropic thermal expansion of the single crystal Bis(pyridinium) tetrachlorocuprate (4) induced by a thermodynamically assisted reorientation of pyridinium cations. **a**, The crystal structures of **3** in the low-temperature phase (left: 123 K; middle: 340 K) and the high-temperature phase (right: 370 K). The pyridinium cations, which statically lean on the adjacent molecules at a tilt angle of 74°, rotationally stand up at the high-temperature phase. The reorientation of the cations induces a large thermal expansion of the single crystal with the directions of principal axes denoted in the left panel. **b**, The anisotropic thermal expansion obtained from *PASCal* program along the principal axes. As the dihedral angle between the pyridine plane and the crystallographic (001) plane shifts from 74° at 123 K to 90° at 373 K, a *ca.* 7% expansion of the crystal along its principal axis was detected. **c**, The DSC curve of crystal. The sharp anomaly peaks and wide thermal hysteresis loop (*ca.* 11 K) indicate the first order phase transition of **4**. **d**, The thermal vibrations of pyridinium at different temperatures. The large thermal ellipsoids of the cations suggest the pyridinium cations are rotationally disordered at a high-temperature phase.

Notably, we recently found that a large but abrupt shape change can be observed in a single crystal of bromide analog, which implies that control of thermal expansion via the thermodynamic rotation-assisted reorientation of the molecular wheel is an effective method. The detailed analyses of this new compound will be performed in a future study.

In the SI:

The deuteration of the sample induces slightly decrease in the phase transition temperatures, 103 suggesting the hydrogen bond interactions in the crystal play minor roles in the phase 104 transitions. A reference for this statement and further elaboration is needed.

Answer: Thank you for the exhaustive review. According to your suggestion, we synthesized a full deuterated analog, bis(*D*₅-imidazolium) tetrachlorocuprate (**3**), where all the H atoms attached on the N atoms and C atoms were replaced with D, to further probe the effect of hydrogen bond interactions on the thermal expansion properties. As shown in Supplementary Fig. 5 (see below), the deuteration of H atoms attached on the C atoms (Compound **2**) induces a slight variation in the phase transition temperatures; however, the full deuteration (Compound **3**) induces a *ca.* 1.5 K decrease in the phase transition temperatures. The larger decrease in the phase transition temperatures of compound **3** is consistent with that the N–H···Cl interactions, which is stronger than the C–H···Cl interactions. Furthermore, we measured the crystal structure of **3** and compared it with compound **1** at each phase. No significant change in the crystal structure was detected after deuteration. We investigated the temperature-dependent cell parameters of compound **3**. As shown in Supplementary Fig. 21, both the cell lengths and cell angles of compound **3** are almost the same as that of compound **1**. The small changes in the phase transition temperatures and similar cell parameters suggest that the deuteration have a little effect on the crystal transformation. We compared the

deuteration effect found in the present compounds with other reported hydrogen-bonded compounds (Ichikawa, M. *et al.*, *J. Mol. Struct.* **1996**, *378*, 17-27; Maczka, M. *et al.*, *Inorg. Chem.* **2014**, *53*, 457-467; Horiuchi, S. *et al.*, *J. Am. Chem. Soc.* **2005**, *127*, 5010-5011; Yao Z.S. *et al.*, *J. Am. Chem. Soc.* **2016**, *138*, 12005-12008.) and found that the shift in phase transition temperatures of present compounds is relatively small. However, we considered it would not be accurate to conclude that *the hydrogen bond interactions in the crystal play minor roles in the phase transitions*. Hence, we modified the related sentence in SI and added several references as presented below:

The deuteration of the sample induces a slight decrease in the phase transition temperatures: *ca.* 0.4 K for compound **2** and *ca.* 1.5 K for compound **3**. These values are relatively smaller than those for other hydrogen-bonded phase transition compounds, suggesting that the deuteration has a little effect on the phase transition temperatures of **1**⁴⁻⁷.

Notably, the result of deuteration was confirmed by comparison between IR infrared spectra of **1** and **3** (Supplementary Fig. 22).

New Figures, Tables, and references were added in the manuscript.

**Supplementary Fig. 5 | Differential scanning calorimetry of 1, 2, and 3.** The two peaks observed in the heating and cooling process indicate that a reversible 2-step phase transition occurs. The deuteration of the sample induces a slight decrease in the phase transition temperatures: *ca.* 0.4 K for compound **2** and *ca.* 1.5 K for compound **3**. These values are relatively smaller than those for other hydrogen-bonded phase transition compounds, suggesting that the deuteration has a little effect on the phase transition temperatures of **1**[4-7]. The broad anomalous peaks and the narrow hysteresis loop (*ca.* 3 K) suggest that the phase transitions occur in a second-order fashion, which is in agreement with the gradual structural transformation observed in the variable-temperature SCXRD analyses.

Supplementary Fig. 21 | Comparison of the temperature-dependent cell parameters of compound 1 and 3. The deuterated analog has very similar cell parameters with that of compound 1.

Supplementary Fig. 22 | IR spectra of bis(imidazolium) tetrachlorocuprate (1) and bis(*D*₅-imidazolium) tetrachlorocuprate (3).

Supplementary Table 3 | Crystallographic parameters for crystal 3.

	LTP	LTP	ITP	HTP
Temperature/K	123	293	373	393
Empirical formula	$C_{12}D_{20}Cl_8Cu_2N_8$			
Formula weigh	707.16			
Crystal size/mm	0.150 × 0.130 × 0.080			
Space group	P -1	P -1	P 2 ₁ / c	Pbam
a /Å	7.7421(12)	7.7577(15)	12.216(2)	7.717(2)
b /Å	11.2979(17)	11.701(2)	7.7205(14)	14.505(4)
c /Å	14.791(2)	14.764(3)	14.545(3)	12.380(3)
α /°	85.111(5)	85.769(7)	90	90
β /°	88.815(5)	89.707(6)	92.377(5)	90
γ /°	86.290(5)	86.973(6)	90	90
Volume/Å ³	1286.2(3)	1334.7(4)	1370.6(4)	1385.8(6)
Z	2	2	2	2
D _{calc} /mg·m ⁻³	1.826	1.760	1.714	1.695
μ /mm ⁻¹	2.502	2.411	2.348	2.322
F ₀₀₀	684	684	684	684
Reflections collected	11455	21454	11890	8596
Independent reflections	5168	7391	2833	1291
R (int)	0.0484	0.0797	0.0577	0.0530
Completeness	98.5%	99.7%	99.6%	99.9%
Data/restraints/parameters	5168/0/271	7391/0/271	2833/0/136	1291/0/73
Goodness-of-fit on F ²	1.131	1.210	1.169	0.643

R_I^a [$I > 2\sigma(I)$]	0.0589	0.0838	0.0959	0.0724
ωR_2^b (all data)	0.1659	0.2079	0.2526	0.2739
^a $R_I = \Sigma F_o - F_c /\Sigma F_o $. ^b $\omega R_2 = \{\Sigma[\omega(F_o^2-F_c^2)^2]/\Sigma[\omega(F_o^2)^2]\}^{1/2}$				

In the methods section:

Bis(imidazolium-*d*₅) tetrachlorocuprate (3). The full deuterated compound **3** was synthesized via the recrystallization of compound **2** twice from methanol-OD solvent under an N₂ atmosphere. Result of deuteration was confirmed by IR spectra and temperature-dependent cell parameters of crystal (Supplementary Figs. 21 and 22). Anal. C₆D₁₀Cl₄CuN₄ (353.5) Calcd C: 20.38, N: 15.84; found C: 20.47, N: 16.10.

Again, we would like to express our gratitude to all reviewers for providing insightful comments with regard to our manuscript.

Reviewers' comments:

Reviewer #1 (Remarks to the Author):

I have evaluated the responses to the referees' comments. I am still strongly of the opinion that the authors CANNOT report thermal expansion coefficients for a temperature range that spans even one (let alone two) phase changes. The authors show that phase changes occur at temperatures of 360 and 387 K, yet report "extremely large" thermal expansion coefficients in the range 293 to 393 K. They then state that these are among the largest values ever observed for solid-state materials. They fail to recognize that, whenever there is a sharp discontinuity in unit cell dimensions (as generally occurs during a phase change), you will obtain very large coefficients. This is especially so if you choose to use unit cell parameters in a narrow temperature range on either side of the phase change. While the authors have not done this, doing so would illustrate my point – i.e. that the numbers become meaningless if not for a single phase.

If everyone in the community were to report thermal expansion coefficients that include phase changes, then the values reported by the authors would appear to be relatively modest in comparison. Therefore their statements about theirs being some of the largest values ever reported would simply fall away. If they want to follow this route, then they should conduct a survey of all materials that show thermosalient effects and "compare apples with apples". To illustrate my point, I have accessed two recent articles on thermosalient phase transitions and this is what I calculated for linear thermal expansion coefficients:

Scientific Reports, 2016, 6, Article number 29610

-3148 -2115 5315 MK-1 in the range 363 to 390 K

Chem. Commun., 2018, 54, 6208:

-4128, -2048 and 6648 MK-1 in the range 160 to 180 K

These authors could have tried to pass these values off as by far the highest ever recorded, but they did not because of the points I have raised above. I do not support publication of this manuscript if the authors insist on following this line of argument. It would set a bad precedent for the field where authors could start cherry-picking data that produce new records that are meaningless.

Reviewer #2 (Remarks to the Author):

I'd like to thank the authors for considering my comments and for making changes to the manuscript.

I am still a little concerned with the analogy of wheels and tops being used to explain the mechanism. There is no up or down in a crystal - so a molecule can't stand up or lie down. It can change orientation in space relative to other molecules but this is not to say it is standing up or lying down.

As I said in my review - the experimental work is very good and I would like to see it published but I think the authors are overstating the certainty of their mechanism and the above analogy is confusing.

When the crystal is heated, all the molecules undergo increased thermal motion. Is it the thermal motion of the anions that causes the cations to alter their relative position in space, e.i. allowing/causing the Hmid cations to change their orientation or is it (as the authors contend) the cations rotating through thermal motion that causes them to change orientation and thus influence the relative positions of the anions? Most likely the change in structure is a consequence of both, rather than just the cations. Certainly, I agree that the relative change in orientation of the cations allows the large simultaneous positive and negative because they are flat - so large changes in metric dimensions of the crystal can occur when they change orientation.

This does not mean that they change orientation solely because they begin to rotate. They could just rotate in the same plane without huge changes to the positions of the anions and thus without the large associated thermal expansion.

I think that the authors should do away with the tops and hoops analogies. I understand that this mechanism has been published elsewhere but as a referee for this NCOMMS paper - I can't concur with it in this case. I am happy for the authors to reference other work as a possible explanation for what is observed here but there needs to be a statement that it is just a possible explanation. The cause and effect is not clear in this case and I think the text of the manuscript should reflect that.

For example, the explanation could read more like this 'The increase in thermal motion of the Hmid on heating appears to cause a change in the relative positions of the cations with consequent change in the orientation of the plane of the Hmid. The result is that the crystal length increases in X direction and decreases in the Y direction. To me that is quite simple and doesn't need analogies to support it.

Reviewer #3 (Remarks to the Author):

The scientific corrections the authors have made to the manuscript are excellent but I continue to be skeptical of the author's generous use of the gyroscope analogy in the text and SI. The analogy continues to feel more like they are trying to make a case that molecules feel such forces and that the rotation is the cause of the molecule standing up because it is trying to minimize gyroscopic forces. This is problematic in the context of molecular solids for two related reasons: (i) there is no "up" in a molecular solid since the relative mass of a molecule makes gravity an irrelevantly small force and (ii) gyroscopic forces on such small molecules will likewise be too small in packed solids to make any difference (extensive use of molecular gyroscopic forces have been discussed by others see work by Garcia-Garibay among others for relative gyroscopic forces in solids). Even if the authors are trying to make a conceptual comparison, it simply lacks the fundamental physical connection. Indeed, the authors site a review by Naumov that looks at **rotor-stator** systems with the exception of the author's own system (Sato), which is included. In this case, the steric argument made in their pentacene paper is a bit out of place with the rest of these systems in Naumov's review but is better supported with sterics causing "loser" packing in the solid than in this case, where sterics clearly can't be called upon to make such a logical leap.

The second issue is that the authors have not demonstrated that the rotation is causative and not merely a coincidence of the orientational change (i.e. correlation does not amount to causation). They have shown that the rotational dynamics increase with temperature and that the molecules reorient such that they are more tightly packed, but they do not provide compelling computational evidence to connect the two.

Given the broad readership of the journal it is important that such analogies not be misinterpreted thus I recommend a straightforward computational experiment that might help: does the molecular rotation favor co-planar stacking between the rings while in a static state the rings prefer to be offset at some angle?

Here they state:

250 non-coplanar intermolecular interactions (Supplementary Fig. 2), i.e., the intensified
251 thermodynamic rotation of molecular wheels upon heating relaxes the restriction of
252 intermolecular interactions; therefore, the molecules can perform a significant
253 reorientation upon the non-planar intermolecular interactions.

Is this statement computationally validated? If it is, this language in the DFT section should be made very, very explicit as this is a fundamentally interesting result. E.g. what (numerically) are the energy differences of the intermolecular interaction energies between static and dynamic states? Is it factually true (supported by theory) that there is a greater repulsive interaction in the low temperature static state that becomes "blurred" upon heating by distributing the charge over a wider area? This seems intuitive but needs proof. If it exists and I have misunderstood, this needs to be more explicit.

Response to reviewer:

Thank you for providing valuable comments to improve the quality and validity of this manuscript. We have revised the manuscript according to the reviewers' comments.

Response to reviewer 1:

Comments: I have evaluated the responses to the referees' comments. I am still strongly of the opinion that the authors CANNOT report thermal expansion coefficients for a temperature range that spans even one (let alone two) phase changes. The authors show that phase changes occur at temperatures of 360 and 387 K, yet report "extremely large" thermal expansion coefficients in the range 293 to 393 K. They then state that these are among the largest values ever observed for solid-state materials. They fail to recognize that, whenever there is a sharp discontinuity in unit cell dimensions (as generally occurs during a phase change), you will obtain very large coefficients. This is especially so if you choose to use unit cell parameters in a narrow temperature range on either side of the phase change. While the authors have not done this, doing so would illustrate my point – i.e. that the numbers become meaningless if not for a single phase.

If everyone in the community were to report thermal expansion coefficients that include phase changes, then the values reported by the authors would appear to be relatively modest in comparison. Therefore their statements about theirs being some of the largest values ever reported would simply fall away. If they want to follow this route, then they should conduct a survey of all materials that show thermosalient effects and "compare apples with apples". To illustrate my point, I have accessed two recent articles on thermosalient phase transitions and this is what I calculated for linear thermal expansion coefficients:

Scientific Reports, 2016, 6, Article number 29610

-3148 -2115 5315 MK-1 in the range 363 to 390 K

Chem. Commun., 2018, 54, 6208:

-4128, -2048 and 6648 MK-1 in the range 160 to 180 K

These authors could have tried to pass these values off as by far the highest ever recorded, but they did not because of the points I have raised above. I do not support publication of this manuscript if the authors insist on following this line of argument. It would set a bad precedent for the field where authors could start cherry-picking data that produce new records that are meaningless.

Answer: Thank you for your detailed comments combined with convincing and useful examples. According to your suggestions, we have recalculated the thermal expansion coefficients only for the low temperature phase (273 K to 353 K). As a result, the linear thermal expansion coefficients change to $\alpha_{a'} = -38 \times 10^{-6} \text{ K}^{-1}$, $\alpha_{b'} = 568 \times 10^{-6} \text{ K}^{-1}$, and $\alpha_{c'} = -184 \times 10^{-6} \text{ K}^{-1}$, and the volumetric thermal expansion coefficient is $346 \times 10^{-6} \text{ K}^{-1}$. These values are smaller than the ones shown in the last submission; however, they remain among the largest values found in solid state materials. Therefore, the conclusion of the manuscript remains unaffected. We have changed the corresponding values in the manuscript as below:

As shown in Fig. 2d and Supplementary Fig. 8, a colossal anisotropic thermal expansion with coefficient values of $\alpha_{a'} = -38 \times 10^{-6} \text{ K}^{-1}$, $\alpha_{b'} = 568 \times 10^{-6} \text{ K}^{-1}$, and $\alpha_{c'} = -184 \times 10^{-6} \text{ K}^{-1}$ was observed in the temperature range from 273 K to 353 K.

Due to the colossal PTE along the principal b' -axis, a large volumetric thermal expansion with a coefficient of $346 \times 10^{-6} \text{ K}^{-1}$ was found in the temperature range from 273 K to 353 K (Supplementary Fig. 9).

Response to reviewer 2:

I'd like to thank the authors for considering my comments and for making changes to the manuscript.

I am still a little concerned with the analogy of wheels and tops being used to explain the mechanism. There is no up or down in a crystal - so a molecule can't stand up or lie down. It can change orientation in space relative to other molecules but this is not to say it is standing up or lying down.

As I said in my review - the experimental work is very good and I would like to see it published but I think the authors are overstating the certainty of their mechanism and the above analogy is confusing.

When the crystal is heated, all the molecules undergo increased thermal motion. Is it the thermal motion of the anions that causes the cations to alter their relative position in space, e.i. allowing/causing the Hmid cations to change their orientation or is it (as the authors contend) the cations rotating through thermal motion that causes them to change orientation and thus influence the relative positions of the anions? Most likely the change in structure is a consequence of both, rather than just the cations. Certainly, I agree that the relative change in orientation of the cations allows the large simultaneous positive and negative because they are flat - so large changes in metric dimensions of the crystal can occur when they change orientation.

This does not mean that they change orientation solely because they begin to rotate. They could just rotate in the same plane without huge changes to the positions of the anions and thus without the large associated thermal expansion.

I think that the authors should do away with the tops and hoops analogies. I understand that this mechanism has been published elsewhere but as a referee for this NCOMMS paper - I

can't concur with it in this case. I am happy for the authors to reference other work as a possible explanation for what is observed here but there needs to be a statement that it is just a possible explanation. The cause and effect is not clear in this case and I think the text of the manuscript should reflect that.

For example, the explanation could read more like this 'The increase in thermal motion of the Hmid on heating appears to cause a change in the relative positions of the cations with consequent change in the orientation of the plane of the Hmid. The result is that the crystal length increases in X direction and decreases in the Y direction. To me that is quite simple and doesn't need analogies to support it.

Answer: Thank you again for your valuable comments. From your detailed explanation, we recognized that the analogy is improper. We have significantly revised the manuscript as follows:

Scheme S1, illustrating the analogy of wheels and tops in the previous revised manuscript (Supplementary Scheme 1. The rotation assists the reorientation in real top and wheel (a) and molecular systems (b)), has been deleted from the Supplementary Information.

All phrases relating to the analogy of wheels and tops have been deleted. These include “stand up”, “lying down”, and “gear racks”. We have also replaced the word “molecular wheel” with (circular) molecular cation or imidazolium throughout the manuscript.

Sentences that include above phrases have been revised, and the changes are directly indicated in the revised manuscript.

As suggested, the structural transformation is a result of synergetic motion of both the molecular cations (Himd^+) and anions, not solely of molecular cations. To explicate this, we added the following sentence containing the phrase “synergetic motion of the molecular cations and the counter anions” in the introduction as follows:

The cation, which is ordered at low temperature, gradually becomes rotationally disordered upon heating. Due to the synergetic motion of the molecular cations and the counter anions, the plane of Himd^+ demonstrates a *ca.* 30° reorientation as the temperature increases from 123 K to 393 K.

We also added the following sentence containing the phrase “synergetic motion of both the cations and anions” in the discussion as follows:

It should be noted that although the motion of Himd^+ cations phenomenologically plays an actuating role, the structural transformation of the material is dominated by synergetic motion of both the cations and anions.

As per the recommendation made in the last part of your comments, we revised the conclusion wherein some sentences directly come your suggestion (“The increase in thermal motion of the Hmid^+ on heating appears to cause a change in the relative positions of the cations with consequent change in the orientation of the plane of the Hmid^+ . The result is that the crystal length increases in X direction and decreases in the Y direction”).

The increase in thermal motion of the Himd^+ cations on heating appears to cause a change in the relative positions of the cations and anions with consequent change in the orientation of

the plane of the Himd^+ . The result is that the crystal length increases in principal b' -axis and decreases in principal a' - and c' -axes.

Response to reviewer 3:

The scientific corrections the authors have made to the manuscript are excellent but I continue to be skeptical of the author's generous use of the gyroscope analogy in the text and SI. The analogy continues to feel more like they are trying to make a case that molecules feel such forces and that the rotation is the cause of the molecule standing up because it is trying to minimize gyroscopic forces. This is problematic in the context of molecular solids for two related reasons: (i) there is no "up" in a molecular solid since the relative mass of a molecule makes gravity an irrelevantly small force and (ii) gyroscopic forces on such small molecules will likewise be too small in packed solids to make any difference (extensive use of molecular gyroscopic forces have been discussed by others see work by Garcia-Garibay among others for relative gyroscopic forces in solids). Even if the authors are trying to make a conceptual comparison, it simply lacks the fundamental physical connection. Indeed, the authors cite a review by Naumov that looks at *rotor-stator* systems with the exception of the author's own system (Sato), which is included. In this case, the steric argument made in their pentacene paper is a bit out of place with the rest of these systems in Naumov's review but is better supported with sterics causing "looser" packing in the solid than in this case, where sterics clearly can't be called upon to make such a logical leap.

Answer: Thank you again for your extremely valuable comment. Based on your and reviewer 2's explanation, we recognize that the analogy and the related structural description in the last submission are improper. Therefore, we deleted the analogy and modified the related description of the mechanism as follows:

Scheme S1, illustrating the analogy of wheels and tops in the previous revised manuscript (Supplementary Scheme 1. The rotation assists the reorientation in real top and wheel (a) and molecular systems (b)), has been deleted from the Supplementary Information.

All phrases relating to the analogy of wheels and tops have been deleted. These include “stand up”, “lying down”, and “gear racks”. We have also replaced the word “molecular wheel” with (circular) molecular cations or Himd^+ throughout the manuscript.

Sentences that include above phrases have been revised, and the changes are directly indicated in the revised manuscript.

The second issue is that the authors have not demonstrated that the rotation is causative and not merely a coincidence of the orientational change (i.e. correlation does not amount to causation). They have shown that the rotational dynamics increase with temperature and that the molecules reorient such that they are more tightly packed, but they do not provide compelling computational evidence to connect the two.

Given the broad readership of the journal it is important that such analogies not be misinterpreted thus I recommend a straightforward computational experiment that might help: does the molecular rotation favor co-planar stacking between the rings while in a static state the rings prefer to be offset at some angle?

Here they state:

250 non-coplanar intermolecular interactions (Supplementary Fig. 2), i.e., the intensified
251 thermodynamic rotation of molecular wheels upon heating relaxes the restriction of
252 intermolecular interactions; therefore, the molecules can perform a significant
253 reorientation upon the non-planar intermolecular interactions.

Is this statement computationally validated? If it is, this language in the DFT section should be made very, very explicit as this is a fundamentally interesting result. E.g. what (numerically) are the energy differences of the intermolecular interaction energies between

static and dynamic states? Is it factually true (supported by theory) that there is a greater repulsive interaction in the low temperature static state that becomes "blurred" upon heating by distributing the charge over a wider area? This seems intuitive but needs proof. If it exists and I have misunderstood, this needs to be more explicit.

Answer: According to your suggestions, we performed detailed theoretical calculations in order to investigate the energy change during phase transition and to illuminate the mechanism of molecular motion. The calculations revealed that the energy in LTP is smaller than in HTP by 5.3 kcal mol⁻¹ per (Himd⁺)₂ [CuCl₄] complex. This is consistent with the experimental observation that the LTP is an enthalpically favorable ground state. Since no distinct geometric change was found in the molecular structures during the phase transitions, the larger energy of the HTP should be predominantly attributed to the reduction in the stabilization energy derived from the intermolecular interactions at the HTP.

Furthermore, the energy barrier that restricts the rotation of Himd⁺ in the molecular plane was significantly reduced from 7.0 kcal mol⁻¹ at the LTP to 3.5 kcal mol⁻¹ at the HTP. The smaller energy barrier is overcome by the energetic wobbling motion of the Himd⁺ cation at the HTP, which is consistent with the observation of the Himd⁺ rotational disorder at the HTP. The presence of multiple available orientations of the Himd⁺ cation contributes to the gain in entropy of HTP.

These computational results validated that the restriction of intermolecular interactions is relaxed at the HTP and the thermodynamic molecular rotation is intensified upon heating. Furthermore, the DFT calculation supports our interpretation that an entropy-driven phase transition occurs upon heating, with the excess enthalpy in the HTP compensated by the entropy gain from intensified molecular motions (a comment from reviewer 3 in the first round of review).

The new calculation results were added in the calculation section of the manuscript as below:

The calculations revealed that the energy of the HTP is higher than that of the LTP by *ca.* 5.3 kcal/mol per (Himd)₂[CuCl₄] complex. The larger energy of the HTP is consistent with the above inference that entropy-driven phase transitions occur upon heating, and suggests intermolecular interactions were reduced in the HTP. Furthermore, the potential energy curve for the rotation of the Himd⁺ cation along the pseudo molecular C₅-axis varies significantly with temperature. As shown in Supplementary Fig. 16, the potential energy barrier for cation rotation decreases from *ca.* 7.0 kcal/mol at the LTP to 3.5 kcal/mol at the HTP. The smaller energy barrier is possibly overcome by the energetic wobbling motion of the Himd⁺ cation.

Furthermore, we calculated the energy change of the repulsive interactions between molecular cations. As we expected, the repulsive energy between a pair of Himd⁺ cations in layer A increases from 69.6 kcal/mol at the LTP to 71.6 kcal/mol at the HTP. The larger repulsive energy between the neighboring Himd⁺ cations suggests that the coplanar packing of Himd⁺ cations at the HTP is unstable and should change to an enthalpy-favorable offset-packing mode at the LTP.

These new calculation results were added in the supporting information (Fig. S18) as below:

The DFT calculations revealed that the repulsive energy between a pair of Himd⁺ cations in layer A increases from 69.6 kcal/mol at the LTP to 71.6 kcal/mol at the HTP, suggesting the co-planar packing of Himd⁺ cations at the HTP is unstable, and should change to an enthalpically favorable offset-packing mode at the LTP.

However, the reorientation of Himd^+ accompanied by the thermodynamic rotation cannot be directly reproduced in theoretical calculations. Therefore, more detailed and intuitive description of the mechanism that can be fully supported by theory is difficult to provide. In order to avoid overstating the mechanism, we modified the discussion in lines 250-253 in previous manuscript as given below:

In the present material, the circular Himd^+ cations act as an entropy reservoir that changes from a statically ordered state into a rotationally disordered state upon heating. The thermodynamic motion of the molecular cations shifts the relative positions of the cations and anions that manifests as collective reorientations of Himd^+ . Correspondingly, a giant linear PTE along the principal b' -axis and NTE along the principal c' -axis were detected.

The detailed calculation method was updated in the supporting information as below:

The rotation energies were calculated by the periodic DFT using the 6.1 version of Quantum ESPRESSO.^{3,4} The widely generalized gradient approximation (GGA) of the Perdew-Burke-Ernzerhof (PBE) functional⁵ and the Blöchl all-electron projector augmented wave (PAW) method⁶ were employed. Plane wave basis sets with a cutoff energy of 500 eV were used for all calculations. Brillouin zone sampling was restricted to the Γ point. The BFGS quasi-Newton algorithm method based on the trust radius procedure was used for geometry optimizations. The climbing-image nudged elastic band (CI-NEB) method⁷ with the quasi-Newton Broyden's second algorithm was used to determine a minimum energy path and to locate a first-order saddle point that corresponds to a transition state. The DFT including the long-range dispersion correction (DFT-D) was also taken into account using the

Grimme semiempirical method⁸ to describe the long-range van der Waals interactions. For all the DFT-D calculations, the energy and force convergence criterions were set as 1×10^{-4} Ry and 1×10^{-3} Ry/Bohr, respectively

3 Giannozzi, P. Baroni, S. Bonini, N. Calandra, M. Car, R. Cavazzoni, C. Ceresoli, D. Chiarotti, G. L. Cococcioni, M. Dabo, I. Dal Corso, A. Fabris, S. Fratesi, G. de Gironcoli, S. Gebauer, R. Gerstmann, U. Gougoussis, C. Kokalj, A. Lazzeri, M. Martin-Samos, L. Marzari, N. Mauri, F. Mazzarello, R. Paolini, S. Pasquarello, A. Paulatto, L. Sbraccia, C. Scandolo, S. Sclauzero, G. Seitsonen, A. P. Smogunov, A. Umari, P. Wentzcovitch, R. M. QUANTUM ESPRESSO: a modular and open-source software project for quantum simulations of materials. *J. Phys.: Condens. Matter* **21**, 395502 (2009).

4 Giannozzi, P. Andreussi, O. Brumme, T. Bunau, O. Buongiorno Nardelli, M. Calandra, M. Car, R. Cavazzoni, C. Ceresoli, D. Cococcioni, M. Colonna, N. Carnimeo, I. Dal Corso, A. de Gironcoli, S. Delugas, P. DiStasio Jr, R. A. Ferretti, A. Floris, A. Fratesi, G. Fugallo, G. Gebauer, R. Gerstmann, U. Giustino, F. Gorni, T. Jia, J. Kawamura, M. Ko, H.-Y. Kokalj, A. Küçükbenli, E. Lazzeri, M. Marsili, M. Marzari, N. Mauri, F. Nguyen, N. L. Nguyen, H.-V. Otero-de-la-Roza, A. Paulatto, L. Poncé, S. Rocca, D. Sabatini, R. Santra, B. Schlipf, M. Seitsonen, A. P. Smogunov, A. Timrov, I. Thonhauser, T. Umari, P. Vast, N. Wu, X. Baroni, S. Advanced capabilities for materials modelling with Quantum ESPRESSO. *J. Phys.: Condens. Matter* **29**, 465901 (2017).

5 Perdew, J. P. Burke, K. Ernzerhof, M. Generalized Gradient Approximation Made Simple. *Phys. Rev. Lett.* **77**, 3865–3868 (1996).

6 Blöchl, P. E. Projector augmented-wave method. *Phys. Rev. B* **50**, 17953-17979 (1994).

- 7 Henkelman, G. Uberuaga, B. P. Jónsson, H. A climbing image nudged elastic band method for finding saddle points and minimum energy paths. *J. Chem. Phys.* **113**, 9901-9904 (2000).
- Frisch, M. J. *et al. Gaussian 09* (Gaussian, Inc., 2009).
- 8 Grimme, S. Semiempirical GGA - type density functional constructed with a long - range dispersion correction. *J. Comput. Chem.* **27**, 1787-1799 (2006).
- 9 Becke, A. D. Density-functional thermochemistry. III. The role of exact exchange. *J. Chem. Phys.* **98**, 5648-5652 (1993).
- 10 Ditchfield, R. Hehre, W. J. Pople, J. A. Self-Consistent Molecular Orbital Methods. 9. Extended Gaussian-type basis for molecular-orbital studies of organic molecules. *J. Chem. Phys.*, **54**, 724-728 (1971).
- 11 Gaussian 09, Revision E01, Frisch, M. J. *et al.* Gaussian, Inc., Wallingford CT, 2016.
- 12 Chai, J.-D. Head-Gordon, M. Systematic optimization of long-range corrected hybrid density functionals. *J. Chem. Phys.*, **128**, 084106 (2008).

As pointed out by reviewer 3 (in the first round of review) and reviewer 2, the phase transitions are a cumulative effect of the entire solid state structure and we should not overemphasize the solely motion of imidazoliums. To clarify this point, we added the following sentence containing the phrase “synergetic motion of the molecular cations and the counter anions” in the introduction as follows:

The cation, which is ordered at low temperature, gradually becomes rotationally disordered upon heating. Due to the synergetic motion of the molecular cations and the counter anions, the plane of Himd^+ demonstrates a *ca.* 30° reorientation as the temperature increases from 123 K to 393 K.

We also added the following sentence containing the phrase “synergetic motion of both the cations and anions” in the discussion as follows:

It should be noted that although the motion of Himd^+ cations phenomenologically plays an actuating role, the structural transformation of the material is dominated by synergetic motion of both the cations and anions.

We would once again like to express our gratitude to all reviewers for providing insightful comments concerning our manuscript.

REVIEWERS' COMMENTS:

Reviewer #1 (Remarks to the Author):

The authors have addressed my comments regarding reporting thermal expansion coefficients across a phase transition.

Reviewer #2 (Remarks to the Author):

The authors have adequately dealt with what I thought were deficiencies in the original manuscript and the again in the revised manuscript.

I think the paper is now suitable for publication in nature comms.

Reviewer #3 (Remarks to the Author):

The authors have addressed the comments to the satisfaction of this reviewer.

Reviewer #1 (Remarks to the Author):

The authors have addressed my comments regarding reporting thermal expansion coefficients across a phase transition.

Reviewer #2 (Remarks to the Author):

The authors have adequately dealt with what I thought were deficiencies in the original manuscript and the again in the revised manuscript.

I think the paper is now suitable for publication in nature comms.

Reviewer #3 (Remarks to the Author):

The authors have addressed the comments to the satisfaction of this reviewer.

Response: We would like to express our sincere gratitude to all reviewers for providing very valuable suggestions to improve the quality and validity of this manuscript.

.